# OpenReview forum: "ABNet: Adaptive explicit-Barrier Net for Safe and Scalable Robot Learning"
_ICML.cc/2025/Conference — ICML 2025 poster_

### Official Review · Reviewer_NgUo · 2025-03-10

**Overall Recommendation:** 3

**Summary:**

This paper proposes ABNet, an adaptive explicit-barrier net for safe and scalable robot learning. The ABNet is a combination of multiple safe control nets such as BarrierNet, dMPC, as well as the proposed explicit-barrier net. The authors claim that ABNet has the potential to scale to a larger safe foundation model and show that ABNet is better in terms of robustness, and safety guarantees over existing approaches.

**Claims And Evidence:**

Yes.
From the experiments, we can see that ABNet outperforms the current existing baselines.

**Essential References Not Discussed:**

The part of references/related works is okay, but it would be nice to have more recent papers included, like the ones published in 2023 and 2024.

**Experimental Designs Or Analyses:**

The experiments are fine for me.

**Methods And Evaluation Criteria:**

Yes.
The explicit-barrier net explicitly computes the optimum control action as the output, which is different from the existing implicit approaches. The experiment setup aligns well with the purposes of the proposed methods. The experiments include 2D navigation and vision-based autonomous driving with obstacles which are common examples of testing control safety. The authors compare their approach with 6 baselines, which should provide enough coverage.

**Other Comments Or Suggestions:**

See above

**Other Strengths And Weaknesses:**

Okay so here comes with the weaknesses.
1. Novelty. The explicit barrier can be seen as a novelty, but it is not significant at all. The way of linear combined control output by different heads is not new. And thus the novelty of the whole paper is concerning.
2. Regarding novelty, as the authors want to push this paper towards a " safe foundation model", I would suggest considering adding a self-attention layer for the outputs by different heads. In this way, even if some heads' output actions are not safe (due to learning errors, etc), the self-attention auto-weighting has the ability to correct the final output.

Strengths:
1. The paper is easy to follow and well-written in general.
2. It is interesting to see the concept of the 'large safe foundation model'.
3. The experiments are well-designed and validate the effectiveness.

**Questions For Authors:**

Are there existing works of 'safe foundation model'?  If yes, has this paper discussed them?

**Relation To Broader Scientific Literature:**

I feel this paper does have its value and contribution to the community, as the authors claimed, towards a large safe foundation model.

The way of considering the combination of multi-safe approaches is interesting to the community.

**Theoretical Claims:**

I've checked the correctness in the main text but not in the appendix. The main text looks good to me.

---

> ### Author Rebuttal · Authors · 2025-03-31
>
> We really appreciate the reviewer for all the positive and helpful comments. We address the remaining comments below.
>
> (1) The part of references/related works is okay, but it would be nice to have more recent papers included, like the ones published in 2023 and 2024.
>
>
> **Response:** We will add more recent references as suggested by other reviewers, especially those in safe RL and CBFs for manipulation.
>
>
> **Weakness 1** Novelty. The explicit barrier can be seen as a novelty, but it is not significant at all. The way of linear combined control output by different heads is not new. And thus the novelty of the whole paper is concerning.
>
>
> **Response:** The proposed explicit barrier is significant in improving the computational efficiency in scalable training, as demonstrated in Fig. 3, we can significantly improve the computation time with our method compared to the benchmark (dQP) from both the batching and training perspectives. Computational efficiency is very important in large safe foundation models, especially when those models are trained in a scalable way (as proposed in this paper).
>
>
> The linear combined control output is simple, but also very effective. However, proving the property (e.g., safety) of the combined control is non-trivial and challenging. The main contribution of our work is to show the safety of the combined control (Thms 3.1 and 3.2, and their proofs in Appendix B), which has not been done in the literature. We showed the existence of a new HOCBF constraint from the combined control, and thus the idea or approach of safety proof is indeed novel.
>
>
> **Weakness 2** Regarding novelty, as the authors want to push this paper towards a " safe foundation model", I would suggest considering adding a self-attention layer for the outputs by different heads. In this way, even if some heads' output actions are not safe (due to learning errors, etc), the self-attention auto-weighting has the ability to correct the final output.
>
>
> **Response:** Thanks a lot for the constructive suggestion. Adding a self-attention layer for the outputs is indeed very interesting. However, proving the safety of the combined control is non-trivial and challenging with such self-attention layers. We will further explore this. One possibility is to consider the linear attention mechanism (e.g., [1] Transformers are rnns: Fast autoregressive transformers with linear attention). We will add the discussion in the revision.
>
> **Question 1** Are there existing works of 'safe foundation model'? If yes, has this paper discussed them?
>
>
> **Response:** We found some survey papers regarding the safety of foundation models (e.g., On the opportunities and risks of foundation models), and we will discuss them in our revision, especially from the perspective of the importance of safety in foundations models, which will make our work stronger. Please let us know if the reviewer has any other suggestions for references.

---

### Official Review · Reviewer_ftwa · 2025-03-11

**Overall Recommendation:** 4

**Summary:**

The paper proposes to embed control barrier constraints into neural layers to enforce safety assurance to network output. In contrast to implicit formulation with differentiable optimization, the paper argues for a specific QP admitting explicit solution form so as to avoid inefficient batching through multi-threading. The explicit barrier layers are duplicated to construct multiple heads, with each accommodating specific safety features, and can be combined via a linear combination. The results show graceful scale-up with respect to the number of batches and network heads comparing to the differentiable optimization counterpart. Imitation learning results on 2D robot navigation, manipulation and vision-based driving show improvement on deriving less conservative safe behaviors.

**Claims And Evidence:**

The main claims about the advantage of explicit barrier QP includes:
1. More efficient inference and batching for training. This is well corroborated with results in Figure 3 in which the proposed approach clearly shows the benefit of avoiding differentiable optimization.
2. Explicit barrier nets improve the learning performance while conform to safety constraints. This is demonstrated in 2D robot navigation and two-link manipulation examples.
3. Multi-head explicit barrier nets are possible due to reduced computational costs and the heads can be combined with each focusing on specific safety-relevant features. This is shown case in the vision-based driving example which also proves the possibility of using unstructured context input z.

**Essential References Not Discussed:**

No extra references are needed.

**Experimental Designs Or Analyses:**

The experiment design covers claims on computational efficiency (Figure 3), safety-assured learning (Table 1, 2, 3), multi-heads to attend different image features (Figure 10) and robustness against image corruption (Table 4). The design is well made and has a good coverage on all the claims made by the paper.

**Methods And Evaluation Criteria:**

The method is based on (Luenberger, 1997) which shows a QP has only two constraints can admit an explicit solution parameterization. To this end, the paper proposes to partition constraints into two sets and choose two that are closest to activeness. The method makes a lot of sense in that explicit parameterization is faster and more amenable to vectored evaluation.

The benchmark includes 2D robot navigation, 2-DOF robot arm and vision-based car driving. The experiments cover different dynamics and vision as contextual input.

**Other Comments Or Suggestions:**

No comments on this regard.

**Other Strengths And Weaknesses:**

The paper needs a clarification on the scope of systems. The part on implicit-barrier (from line 125) suggests the target dynamics is of a general control affine form $\dot{x} = f(x) + g(x) u$ while an assumption is made on relative degree $m$. The assumption is not made in the neighbourhood of some $\bar{x}$ to ensure $g(\bar{x}) \neq 0$ for the relative degree $m$. In general, I think $g(x)$ may not guarantee the same relative degree everywhere in the state space. The experiments do not contain such a case as the dynamics are either fully-actuated or with a constant control matrix. It is hard to tell what if the dynamics violate this relative-degree or when $m$ is misspecified for some states.

The paper advocates scalable learning while the examples are still limited to low-dimensional system and action space. I guess the scalability statement here is about the computational cost of applying differentiable optimization on problems of such a scale. However, I think the contribution can be much more significant if results can be attained on systems with higher DOFs.

The paper seems to take visual observations as piece-wise constant input. This reads as a strong assumption by breaking the causality between state and generated observation, while it is understandable this can save the analysis of differentiating the observation model.

**Questions For Authors:**

1. Can the method be demonstrated to work on dynamics without a globally constant relative degree? How much it may invalidate the safety assurance?

2. The minimum trick to select "the most active" constraints seems general. What could be its implication to other applications? Do we need some care on how to partition the constraints when the problem scales up? I am wondering whether it can work for optimizing along a trajectory where state may change the activeness of constraints and hence the constraints considered in QPs. Will this create in differentiability or complex loss landscape to gradient-based optimization?

3. Differentiable optimization sometimes suffer from infeasible problem parameterization. Will that also be an issue here?

4. The vision-based driving example takes image as the input and generate control from the ABNets (line 813-814). How is this done without the state input? Is the demonstrated safety formally assured or an empirical verification in this specific task?


##update after the rebuttal
I would like to thank the authors' responses and clarifications, especially on the relative degree. Two points I would like to make after thinking over the rebuttals:

- It is good to know that there are already potential "patching" for the cases beyond constant relative degree as in the experiments. I see some issues on resorting to these patches to counter the original criticism. Being able to define safety constraints to let robot operate in a subspace with constant relative degree is not a direct resolution to the criticism on the rigour of the theorem statement. The original statement suggests a fixed relative degree which implicitly assumes this applies to all control matrices. Providing solutions about what we could do if the (implicit) assumption didn't hold is not helping on the clarity and rigorousness of the problem scope. I am also unsure about the implication to empirical performance if state constraints are imposed for making the analysis valid in a subspace. Many under actuated tasks actually rely on going through uncontrollable states and exploiting passive dynamics.

- The clarifications on the vision-based driving and state prediction model make sense. However, ignoring the causal relation between the observation model (state -> image) appears to break the analysis chain of the control loop. This could be fine for empirical driven results, but I feel it is a bit confusing for a work promoting provable control as I thought the example was intended to show the proof also applies to system with unstructured sensory observation.

Overall, I think the paper should still have a good chance to be accepted while I have to admit my confidence is not as strong as it was before the rebuttal, given the way of how the raised questions were approached. My recommendation remains unchanged.

**Relation To Broader Scientific Literature:**

The contribution is related to broader literature on using implicit function for layer design. The findings on effective learning of structured output are in line with existing works arguing for differentiable computing as building blocks for end-to-end learning. The idea of leveraging explicit solution parameterization can be inspiring to works beyond safe learning.

**Theoretical Claims:**

The paper provides the proof on safety assurance of blended network output subject to control barrier constraints in (3). The correctness is briefly checked but not thorough as the reviewer is not familiar with the adaptive CBF cited from literature.

---

> ### Author Rebuttal · Authors · 2025-03-31
>
> We thank the reviewer for all the positive and constructive comments. We address the remaining comments below.
>
> (1) The paper needs a clarification on the scope of systems. The target dynamics is of a general control affine form while an assumption is made on relative degree m...
>
> **Response:** The proposed method can be applied to general nonlinear systems. The problem that the coefficient of the control in the CBF constraint becomes 0 at some state is the so-called ``singularity’’ problem (if it is zero for all states, then the relative degree should be increased by one until it is non-zero). This problem can be solved by defining a CBF that avoids those states (e.g., see [1] High-order barrier functions: Robustness, safety, and performance-critical control). When there are multiple controls, there may be the so-called mixed relative degree problem, and we can address this by defining auxiliary dynamics to make all the controls show up (e.g., see [2] Control barrier functions for systems with multiple control inputs). When we have non-affine control systems, we can also define auxiliary dynamics to ensure that the method can still work (e.g., see [3] On the forward invariance of neural ODEs). We will make this clear in the revision.
>
> (2) Limited to low-dimensional system and action space...
>
> **Response:** We totally agree with the reviewer. The scalability is mainly in terms of the safety of the robot, the size of the safe robot learning models, and the computational cost of the model. Our method can also work for high DOF scenarios (e.g., manipulation) by replacing the corresponding system dynamics with the high DOF dynamics (e.g., manipulator dynamics) in the model. We will discuss this in the revision.
>
> (3) The paper seems to take visual observations as piece-wise constant input, ...
>
> **Response:** We thank the reviewer for pointing this out. Although we are taking piece-wise constant input for visual observations, the proposed model works for continuous visual observation as well. This is indeed a very interesting direction that involves differentiating the image input (e.g., in the form of optical flow). We will add this discussion in the revision.
>
> **Question 1**
>
> **Response:** Our method works for dynamics without a globally constant relative degree as well. There are many approaches to deal with such problems, e.g., by defining a CBF that avoids those states (e.g., see [1] High-order barrier functions: Robustness, safety, and performance-critical control). In fact, we only need to care about the states at the boundary of the unsafe sets that may make the coefficient of control in HOCBFs become zero, and safety can still be preserved in such cases.
>
> **Question 2**
>
> **Response:** The implication to other applications is that we always consider the most threatening factors in guaranteeing safety, and it may not necessarily just  be related to the activeness of constraints, but also related to the importance of those constraints. For example, autonomous vehicles should always follow traffic rules, and it should always consider the most important rule (e.g., ensure collision free to pedestrians) rather than less important rules (e.g., lane or road keeping). We do need to care on how to partition the constraints when the problem scales up, and this can be taken care of by leveraging other factors (such as ethics and local culture in driving), e.g., see [1] Liability, Ethics, and Culture-Aware Behavior Specification using Rulebooks  [2] Rule-based optimal control for autonomous driving.
>
> Since the model outputs controls in real time, we can definitely change the activeness of constraints in real time as well. We implement the model in discrete time, thus it won’t cause any differentiability problem since we can address the inter-sampling issue (safety in continuous time) using event-triggered approaches (e.g., [3] Event-triggered control for safety-critical systems with unknown dynamics). Another way to address this differentiability problem is to use the soft min approach (as discussed right before equation (4)) to combine and consider all the constraints.
>
> **Question 3**
>
> **Response:** This is really a good point. Since we always consider the two most important constraints at each time step, and we can get the closed-form solution, therefore, we do not find any infeasibility issue for now (however, there may be other problems that are worth further exploring in future work).  This provides a promising way to address the infeasibility problem, and we will add the discussion in our revision.
>
> **Question 4**
>
> **Response:** We also have some state- net to predict the vehicle and obstacle states from image observations (as also demonstrated in the BNet). The demonstrated safety is based on the assumption that the state is reliably predicted. In cases where there are some uncertainties for the prediction of the states, we can use robust CBFs in our framework. We will add the discussion in the revision.

---

### Official Review · Reviewer_kxSJ · 2025-03-14

**Overall Recommendation:** 2

**Summary:**

This paper addresses a critical challenge in AI-enabled robotics—safe learning—by introducing the Adaptive explicit-Barrier Net (ABNet). The authors highlight the limitations of existing safe learning methods, including poor scalability, inefficiency, and instability under noisy inputs. ABNet overcomes these issues by explicitly incorporating safety barriers into a closed-form model, ensuring provable safety guarantees. A key innovation is its multi-head structure, allowing different model heads to learn safe control policies from distinct features, thereby improving training efficiency and stability without requiring a monolithic large model. The approach is validated across diverse robotic tasks, including 2D obstacle avoidance, safe manipulation, and vision-based autonomous driving, demonstrating superior robustness and safety compared to existing models. The paper’s contributions are significant in both theoretical and practical aspects, providing a promising direction for scaling safe learning toward foundation models for robotics.

**Claims And Evidence:**

I believe that some of the authors’ claims are not well-supported by the experiments, and the writing of this paper is not particularly clear, making it somewhat difficult to understand. Similar to Control Barrier Functions (CBF), this work primarily addresses the problem of safe robot learning. However, one major concern is the fundamental distinction between the problem studied in this paper and that of Safe Reinforcement Learning (Safe RL). Safe RL also frequently employs Barrier Functions to handle safety constraints, as seen in works such as:

1.	Penalized Proximal Policy Optimization for Safe Reinforcement Learning

2.	IPO: Interior-point Policy Optimization under Constraints

**Essential References Not Discussed:**

some related work is not discussed in the paper.

**Experimental Designs Or Analyses:**

Yes, some concerning see above.

**Methods And Evaluation Criteria:**

Furthermore, the complexity of the experimental environments used in this paper is not well-articulated, making it difficult to assess the significance of the results. Could the authors provide further clarification on the environments used in the study?

**Other Comments Or Suggestions:**

See above.

**Other Strengths And Weaknesses:**

What are the key advantages of using Barrier Functions in the proposed approach?

Moreover, the applicability of this method seems to be constrained by the requirement for known and differentiable constraint functions. In many real-world scenarios, defining such functions can be highly complex, often resembling a black-box system. Real-world uncertainties, noise, and sensor errors may further degrade the algorithm’s performance, potentially leading to failures.

For instance, ensuring that large language models (LLMs) are safely aligned—such as preventing the generation of harmful content—requires defining constraint functions for “human safety,” which is itself a highly challenging task. How do the authors view this issue in the context of their approach?

**Questions For Authors:**

See above.

**Relation To Broader Scientific Literature:**

Additionally, does this method remain effective in environments requiring complex contact handling? Safe robot learning often considers constrained environments, such as those found in safety-gymnasium. The reviewer is interested in understanding how the proposed algorithm performs in more complex scenarios.

The paper’s structure may make it particularly difficult for readers to follow, especially given the lack of clarity in the Background and Related Work sections.

Safe learning has been extensively explored in the context of Safe Reinforcement Learning, with optimization approaches based on Barrier Functions and Lagrangian formulations. Compared to the following works:

1.	Reward Constrained Policy Optimization

2.	Constrained Policy Optimization

**Theoretical Claims:**

Yes. I see the appendix and the main page.

---

> ### Author Rebuttal · Authors · 2025-03-31
>
> We appreciate the reviewer for all the helpful and constructive comments. We address all the concerns below.
>
> (1) Claims are not well-supported by experiments, writing is not particularly clear.
>
> **Response:** Our main claim is safety guarantee of the model, and this is supported by the SAFETY or CRASH  in Tables 1-4. The computational efficiency of our method is demonstrated in Fig. 3 with comparison to benchmarks. The performance improvement in scalable learning is shown in Figs. 4-6. We will improve the writing with preliminary on CBFs and BNet in the revision.
>
> (2) Fundamental distinction with Safe RL and optimization approaches based on BFs and Lagrangian
>
> **Response:** There is indeed a fundamental distinction between our method and Safe RL. In safe RL, safety is usually taken as a component of reward function, and thus, it can only improve safety without guarantees. While our method can formally prove safety, as shown in Thm. 3.1 and 3.2. Safety guarantees are also demonstrated in Tables 1-4. We will add discussions on safe RL and include the references in the revision.
>
> (3) Complexity of environments not well-articulated. Could the authors provide clarification on the environments?
>
> **Response:** The complexity of the experimental environments is given in Appendix section C. In summary, we consider different nonlinear dynamics and constraints across three different tasks. We have one complex vision-based end-to-end autonomous driving experiment in which the model directly takes the front-view image as input, and outputs the safe control. All the experiments are done in VISTA, and it is a sim-to-real driving simulator that generates driving scenarios from real driving data [Amini, et.al 2022].  The VISTA allows us to train model with guided policy learning. This learning method has been shown to work for model transfer to a full-scale real vehicle. This is also given in Appendix section C.4.
>
> (4) Does this method remain effective in environments requiring complex contact handling?
>
> **Response:** Our method is still effective in environments requiring complex contact handling, in which case we just need to replace the dynamics by the ones of manipulator. The safety may be different in contact scenarios (e.g., force constraint instead of collision avoidance). The CBF method has been widely used in manipulation in complex environments (e.g., Safe Multi-Robotic Arm Interaction via 3D Convex Shapes), since our model is based on the CBF method, we can apply our model to those complex scenarios when learning is involved.
>
> (5) Lack of clarity in the Background and Related Work.
>
> **Response:** We will add preliminaries on CBFs and BNet to improve the clarity, and discuss more on safe RL and other related works  in Related Works section (now the related work is given in Sec. 5).
>
>
> (6) Some related work is not discussed in the paper.
>
> **Response:** We will add the references suggested by the reviewer and other reviewers (e.g., safe RL), and discuss them.
>
> (7) What are the key advantages of using Barrier Functions in the proposed approach?
>
> **Response:**  The key advantages of using barrier functions in the proposed method is that we can formally train models that have safety guarantees in a scalable way, as shown in Thms. 3.1 and 3.2 and Tables 1-4. Another advantage is the high efficiency of our proposed method, as shown in Fig. 3.
>
> (8) Applicability constrained by known and differentiable constraint functions. Uncertainties, noise, and sensor errors may further degrade performance, potentially leading to failures.
>
> **Response:** There are many related works that learn differentiable constraint functions from demonstrations, such as [1] Learning control barrier functions from expert demonstrations and [2] Synthesis of control barrier functions using a supervised machine learning approach. Therefore, we can combine existing literature with our ABNet to learn constraints in scenarios that the safety is not predefined. This has been made clear in the future work section 6. When there are uncertainties, noise or sensor errors, we can employ robust CBF methods in our model (e.g., Fault tolerant neural control barrier functions for robotic systems under sensor faults and attacks). We will make this clear in the revision.
>
>
> (9) large language models (LLMs) (safely aligned): preventing harmful content—requires defining constraint functions for “human safety,” How do the authors view this issue in the context of their approach?
>
> **Response:** Thanks a lot for pointing out the safety problem of LLMs, we have been actively working on this and on other models or problems where safety is not clearly defined. In LLMs, in order to avoid toxic language, we may learn a differentiable risk function according to the harmfulness of the content, and then we can take this risk function as a CBF in our approach such that we ensure that the harmfulness of generated content be below some level. We will discuss this in the revision.

---

> > ### Comment · Reviewer_kxSJ · 2025-04-08
> >
> > Thank you very much for the authors’ response.
> >
> > Over the past two days, I have revisited the manuscript and the rebuttal while also reviewing the comments from other reviewers.
> >
> > Thank you for the reviewer’s response. I have decided to keep my score unchanged.
> >
> > I sincerely appreciate the efforts the authors have made during the rebuttal period.

---

> > > ### Author Response · Authors · 2025-04-08
> > >
> > > We really appreciate the reviewer for the comment.
> > >
> > > Could you please let us know the reasons of keeping the score unchanged? Any point would help a lot to further improve our paper in the revision. Please let us know if you have any remaining concerns.
> > >
> > > Thank you,
> > >
> > > Authors.

---

### Official Review · Reviewer_KxjS · 2025-03-16

**Overall Recommendation:** 3

**Summary:**

The paper presents ABNet, a novel framework that utilizes attention mechanisms to handle diverse input patterns, while incorporating barrier functions to maintain the system state within a safety set, ensuring forward invariance. This approach aims to improve the scalability and robustness of robot learning by enabling each head of BarrierNet to focus on different aspects of the observation space, thereby facilitating the development of safe control policies in a variety of environments.

**Claims And Evidence:**

While the incorporation of attention mechanisms into BarrierNet represents a core contribution, the manuscript would benefit from a more thorough technical exposition regarding the non-trivial nature of this integration. The authors should elaborate on: (1) specific technical challenges encountered during this integration process, (2) innovative solutions developed to overcome these challenges, and (3) the distinctive advantages conferred by this particular implementation of attention mechanisms. Such technical insights would significantly enhance our understanding of the methodological novelty and provide clearer differentiation from conventional architectural adaptations.

**Essential References Not Discussed:**

[1] CARLA: An open urban driving simulator

[2] Nuscenes: A multimodal dataset for autonomous driving

**Experimental Designs Or Analyses:**

Figure 6 does not clearly highlight the performance differences between ABNet and methods like MPC or BNet. While Figure 6 includes results from MPC, Table 3 does not provide a corresponding quantitative comparison. For the method to be applicable to more complex, real-world scenarios—such as autonomous driving with dynamic obstacles like vehicles or pedestrians—consideration of the environment's external dynamics is essential. This raises concerns about the scalability of ABNet. The authors could consider testing in more advanced simulators, such as [1-2], to better showcase the method's adaptability and robustness in dynamic environments.

[1] CARLA: An open urban driving simulator

[2] Nuscenes: A multimodal dataset for autonomous driving

**Methods And Evaluation Criteria:**

The experimental scenarios are relatively simplistic, consisting of static environments with limited dynamics, and lack sufficient qualitative analysis, such as video demonstrations comparing the method to baseline approaches.

**Other Comments Or Suggestions:**

To my knowledge, large transformer models are increasingly applied in autonomous driving scenarios involving dynamic external agents, as referenced in [3]. This approach uses transformers as a backbone for safety planning based on sampled trajectories, ensuring collision-free paths while closely resembling NMPC planners or human driving behaviors. How do the authors view the online planning, and could barrier functions be integrated to enhance safety?

[3] Planning-oriented Autonomous Driving

**Other Strengths And Weaknesses:**

1. The approach explicitly incorporates barrier functions into neural network training, ensuring safety constraints are satisfied.
2. The modular architecture of ABNet with multiple attention heads allows scalable, incremental learning, which is promising for building complex, safe models in stages.
3. The method demonstrates robustness to noise, yielding lower variance in performance.
4. The method's practical application is constrained by its dependence on predefined, differentiable constraint functions. This requirement poses significant challenges in real-world implementations, where such functions are often either prohibitively complex to formulate or behave as essentially unknowable systems.

**Questions For Authors:**

Can the method handle complex contact scenarios and adapt to humanoid robots managing full-body dynamics with external force feedback?

**Relation To Broader Scientific Literature:**

The proposed methodology may prove particularly suitable for safety-critical applications with well-characterized system dynamics.

**Theoretical Claims:**

The method does not appear to ensure optimal task performance. As stated in the paper, "we use NMPC to collect ground-truth controls (training labels) with corresponding states," implying that the upper limit of ABNet's task performance is constrained by the performance of NMPC (e.g., minimum time to reach a target). Additionally, optimality does not seem to be the primary focus in training. By employing imitation learning with barrier functions, safety appears to be prioritized, which could further impact task performance. Is there a mechanism to balance task performance and safety? Furthermore, the criteria for selecting the penalty functions in Equation 3 are not well explained. How are these values chosen, and how does the method balance conservatism and performance optimality?

---

> ### Author Rebuttal · Authors · 2025-03-31
>
> We appreciate the reviewer for all the positive and constructive comments.
>
> (1) Specific technical challenges
>
> **Response:** There are two main technical challenges: (a) The training and testing efficiency of the scalable robot learning model; (b) Formal proof for the safety of the composed model in scalable robot learning;
>
> (2) Innovative solutions to overcome these challenges
>
> **Response:** For challenge (a), we proposed the explicit-barrier approach (in which a closed-form solution for the QP is given)  to significantly improve the efficiency of the model, as shown in Fig. 3. For (b), we show the existence of a new HOCBF condition from the composed model, which has not been done in the literature ((21) of Appendix sec. B).
>
> (3) Distinctive advantages
>
> **Response:** The most distinctive advantages of the method is the safety guarantee, as shown in Tables 1-4 (SAFETY). This is also the main contribution of our work. We also show the improvement of performance with the increase of heads of explicit-barrier, as given in Figs. 4-6 and Table 3 (PASS).
>
>
> (4) Experiment are relatively simplistic, and lack qualitative analysis.
>
> **Response:** We have two intuitive experiments that are easy to understand (from dynamics, safety perspective). We also have one complex vision-based end-to-end experiment to show how our methods can work in realistic scenarios. We have shown qualitative analysis in Fig. 4, Fig. 8, 9, 11.  All the qualitative results are from videos, and we will attach them in the paper revision (the rebuttal only allows pictures).
>
> (5) Optimal performance: Is there a mechanism to balance performance and safety?
>
> **Response:** The objective of this paper is to show the safety instead of optimality. However, as we solve dQPs, we can indeed ensure optimal task performance. For applications where safety is not critical, we can relax the first condition in (3) with a slack variable, and minimize it in the cost (2). In this way, we can balance task performance and safety. We will add this in the revision.
>
> (6) Criteria for selecting penalty functions: how does the method balance conservatism and optimality?
>
> **Response:**  The penalty functions in (3) are output of previous NNs or trainable parameters, and we do not need to select them by hand. Smaller penalty functions will make the robot more conservative. Our method finds optimal functions through training of the model to achieve a desired balance between conservatism and performance (given by the training data). We will make it clear in the revision.
>
> (7) Figure 6: performance differences between ABNet and methods like MPC or BNet. Table 3 does not provide a comparison with MPC.
>
> **Response:** There is a clear improvement of ABNet over BNet as trajectories with BNet stop near the obstacle (shown by blue trajectories in Fig. 6), while the ones of ABNet can safely pass obstacle. The MPC is the ground truth (that is computationally expensive, cannot be applied in real time), and thus it is not included in Table 3. We will make this clear in the revision.
>
>
>  (8) Scalability to dynamic obstacles, and testing in more advanced simulators.
>
> **Response:** Our ABNet can be scaled to dynamic environments as we just need to incorporate the dynamics and states of obstacles in (3). However, identifying dynamics and states of obstacles requires more powerful models, which does not rely on ABNet (not a limitation of ABNet). The ABNet mainly provides safety guarantees, and we can use existing literature in dynamics and state estimation to better augment ABNet in dynamic environments.  We will explore it in future, and make it clear in the revision.
>
>
> **Weakness:** Dependence on predefined, differentiable functions.
>
> **Response:** There are many related works learning differentiable constraints from demonstrations, e.g., [1] Learning control barrier functions from expert demonstrations, [2] Synthesis of control barrier functions using a supervised machine learning approach. Therefore, we can combine existing literature with ABNet to learn constraints in cases that safety is not predefined.
>
>
> **Suggestion:** Large transformer models with safety planning. How do the authors view online planning, and could barrier functions be integrated to enhance safety?
>
> **Response:** Existing approaches with large models can improve safety, however, there are no guarantees. Our method can be integrated into them to formally guarantee safety. The transformer models can be integrated either in the upstream or downstream of ABNet (e.g., via linear attention such that we can still prove the safety of the model).
>
> **Question:** Can the method handle complex contact scenarios?
>
> **Response:** Our method can handle complex contact scenarios by replacing the dynamics with the ones of humanoid. The safety may be different in contact scenarios (e.g., force constraint instead of collision avoidance). This would be another important application of our model, and we will discuss it in the revision.

---

> > ### Comment · Reviewer_KxjS · 2025-04-06
> >
> > My concerns remain unresolved. In the experimental environment provided by the authors, all obstacles are static and two-dimensional. However, the authors still claim that MPC serves as a computationally expensive ground-truth method. I believe that in such a simple environment, MPC results should be included.
> >
> > The authors mention that the penalty function is the output of a neural network, which I find confusing. How is this penalty function trained? This aspect is not discussed in the paper. Does this imply that additional training data is required to learn the penalty function?

---

> > > ### Author Response · Authors · 2025-04-06
> > >
> > > We appreciate the reviewer for the further feedback.
> > >
> > > **1. MPC results should be included.**
> > >
> > > **Response:** We have compared MPC with our ABNet in the vision-based end-to-end autonomous driving task, please see the table below. In summary, both methods can make the ego vehicle safely pass the obstacle (as also shown by the Safety metric that should be $\geq 0$ for a safe control method). However, the MPC (with interior point to solve the nonlinear programs) is much more computationally expensive than our proposed ABNet (0.872s v.s. 0.004s at each time step). Since the MPC involves the solving of nonlinear programs, the solver usually has to linearize or simplify the model within itself to increase the computational efficiency, which may make MPC lose safety guarantees. Moreover, we found that MPC sometimes gives very awful solutions that are far from optimal (due to the complexity of the nonlinear dynamics and constraints). While our ABNet can transform nonlinear optimizations into differentiable QPs, and then further derive the closed-form solution with the proposed method. ABNet can always give normal solutions with theoretical safety guarantees. We will add the results to the revision of the paper.
> > >
> > >
> > > |  Method| Crash| Pass|Safety|Computation time| Theoret. Guar.|
> > > |---|---|---|---|---|---|
> > > |  MPC|0% | 100% | 0.006 | 0.872| $\times$|
> > > |  ABNet|0% | 100% |1.455 | 0.004| $\surd$|
> > >
> > > **2. The authors mention that the penalty function is the output of a neural network, which I find confusing. How is this penalty function trained? This aspect is not discussed in the paper. Does this imply that additional training data is required to learn the penalty function?**
> > >
> > > **Response:** Please note that the penalty function is not the output of the ABNet. The only output of the ABNet is the solution of the dQP/closed-form solution (i.e., the control of the robots). Instead, the penalty function is the input of the ABNet (**please note that this is clearly shown by the inputs $p_i$ and $p_{m,1}$ in Fig. 2**), and it is the output of the previous layer (e.g., LSTM in the vision-based driving case study).
> > >
> > > Therefore, we can just take penalty functions as some intermediate variables (or parameters) within the neural network (just like other trainable parameters of a neural network), and we do not need any additional data to train the penalty function. However, we do require the penalty function to be positive in order to ensure the safety in the ABNet, and we have used a scaled sigmoid function to ensure the penalty function to be $> 0$.
> > >
> > > In summary, the penalty function is just like other trainable parameters in a neural network, and it is optimized by the loss of the output of the ABNet (controls of a robot) using error backpropagation. **We train the ABNet like a normal neural network using error backpropagation to optimize all the parameters (including the penalty function)**.

---

### Decision · Program_Chairs · 2025-05-01

**Decision:**

Accept (poster)

**Comment:**

Based on my review and the authors' rebuttal, I recommend accepting this paper. The core contribution, an explicit control barrier layer derived from an efficiently solvable QP, presents a valuable alternative to implicit differentiable optimization for enforcing safety constraints in neural networks. The authors provide convincing evidence through well-designed experiments across navigation, manipulation, and vision-based driving tasks, demonstrating significant improvements in computational efficiency, scalability with batch sizes and network heads, and the ability to learn less conservative yet safe behaviors. While I noted limitations regarding the theoretical assumptions on system dynamics (specifically the constant relative degree) and the handling of causality in the vision-based example, which were not fully resolved in the rebuttal, the practical benefits and the novelty of leveraging explicit solution parameterizations for structured output prediction in deep learning are significant. Therefore, despite some reservations about the theoretical scope and rigor that arose post-rebuttal, I believe the paper's strengths and potential impact warrant its acceptance.